# Planning for Urban Social Sustainability: Towards a Human-Centred Operational Approach

**Céline Janssen \*** , **Tom A. Daamen and Co Verdaas**

Management in the Built Environment, Delft University of Technology, 2628 BL Delft, The Netherlands;
t.a.daamen@tudelft.nl (T.A.D.); j.c.verdaas@tudelft.nl (C.V.)
**\*** Correspondence: celine.janssen@tudelft.nl

**Abstract:** In Europe, growing concerns about social segregation and social stability have pushed calls to make cities 'inclusive, safe, resilient and sustainable' higher on policy agendas. However, how to approach such generic policy objectives and operationalise them for planning practices is still largely unclear. This article makes a conceptual contribution to the operational understanding of social sustainability in urban planning practices. The article argues that, between theoretical concept and operational forms, different evaluative approaches towards social sustainability may be taken. Evaluating three dimensions of policy operationalisations in The Netherlands, we argue that Amartya Sen's capability approach provides a promising conceptual framework for operationalising social sustainability in cities in Europe and beyond. We compare capabilities with a more commonly applied resource-based conception to show that the former is more accurate and potentially more effective, because it shifts the evaluative space of social sustainability from means (i.e., urban resources) to ends: the eventual well-being of urban citizens.

**Keywords:** social sustainability; operationalisation; capability approach; urban planning practices; The Netherlands

## 1. Introduction

While much attention has been accorded to the economic and environmental aspects of sustainability in cities, the social dimension of urban sustainability has recently also started to receive its share of scrutiny in both research and practice. In Europe, concerns about social segregation and social stability in member states have led the European Committee [1] to pledge for making cities more secure places to live by emphatically adopting the United Nation's sustainable development goals [2]. The new policy objectives—for example on urban inclusiveness, safety and resilience—have accelerated efforts in academic and professional networks to better understand how the these may be translated into context-specific approaches and operationalisations (e.g., EUROCITIES [3] and ULI [4]).

Not only do European cities face socio-economic challenges, such as increasing spatial segregation between income groups [5] and increasing economic inequality [6], they are also confronted with several social imbalances. In the Netherlands, for instance, citizens experience stronger tensions between 'the rich and the poor', and researchers have observed an increase in conflict between Dutch natives and people with a migration background [7]. Similar to other places in Europe, it is found that citizens more frequently express feelings of societal unease, and that polarization, 'hardening' and radicalisation are lurking [8]. The need for policy makers to address these issues has thus been mounting.

Although a vast number of studies on social sustainability in the built environment have emerged in recent decades (see e.g., Manzi et al. [9] and Colantonio and Dixon [10]), only few have focused on how social sustainability, as a policy goal, might be operationalised in planning practices. Some authors claim that "despite the overall consensus about the significance of social sustainability in the sustainable development agenda, a

common agreement on the definition and operationalisation of this concept is still missing" [11] (p. 623). But what this fails to consider is that perhaps the impossibility of finding such common ground is the very reason that few researchers have offered it. In contrast, we therefore assume that social sustainability, generally defined as "maintaining or improving the well-being of people in this and future generations" [12] (pp. 224–245), is an inherently pluralistic concept. This means that, while acknowledging the importance of the general concept anywhere, a wide range of possibly conflicting operationalisations may be both warranted and empirically sound, given their specific contexts. Therefore, the aim of this article is threefold.

First, we aim to make a conceptual contribution to the operationalisation of social sustainability goals in cities by arguing, following Moroni [13], that different conceptions to social sustainability in research and practice are inevitable, and that these logically result in different operational approaches. Second, we argue that applying a capability-based conception, based on the key tenets of Amartya Sen's [14] capability approach, is a more comprehensive approach to social sustainability than what is currently common in urban research and practice. A capability-based evaluative approach takes human and contextual diversity into account and, therefore, draws on a richer informational basis that is particularly helpful for conceptions of social sustainability that focus on human well-being-issues. Thirdly, after explaining our conceptual arguments, we will empirically explore three recent urban planning examples in The Netherlands. Assessing Dutch national policy programmes for urban renewal, national regulations on the country's acclaimed social housing system, and a recent national measurement tool on liveability, we show that Dutch urban policy-making has mainly concentrated on physical interventions that merely address the tangible aspects of social sustainability and, thus, largely miss the crucial intangible dimension of the concept.

The next section elaborates on the conceptual understanding of social sustainability and its operationalisation in planning research or practice. Section 3 continues on the advantages of the capability approach as a new, alternative conception to social sustainability. Section 4 introduces recent Dutch cases of social sustainability operationalisation in planning practice. Section 5 discusses the empirical results considering our capability-conception of social sustainability. The article concludes with the hypothesis that applying a new, capability-based evaluative approach will be able to address social sustainability in a more comprehensive and effective way than currently common in practice.

## 2. Steps of Operationalisation: From a Value-Laden Plurality to Concrete Indicators

The growing number of studies on social sustainability have not led to a single definition of it, but rather to a comprehensive scrutiny of the values, principles and indicators of what social sustainability is about [15–17]. As Shirazi and Keivani [18] (p. 1539) identify, "different approaches to social sustainability have resulted in a fragmented, sometimes contradictory, body of literature." Although some scholars warn that social sustainability, without including the key issue of social justice, is merely a container concept [19], others explain that social sustainability can be seen as "a conceptual tool that policy makers and practitioners can use to communicate, make decisions, and measure or assess current developments, and that scholars can very well study and even refine" [20] (p. 2).

Dempsey et al. [21] suggest that social sustainability is, in essence, about social equity and sustainability of community; Weingartner and Mobert [22] mention social capital, human capital and well-being as central values for social sustainability; Rashid-farokhi et al. [23] conclude on six fundamental values, namely, equity, social inclusion, social cohesion, social capital, community participation and safety. Reflecting on thirty years of research on social sustainability, Shirazi and Keivani observe that social sustainability is "neither absolute nor fixed" [18] (p. 1532), and that scholars simply use different meanings and indicators. Based on a metaliterature review, they list seven principles and key aspects that are commonly used to define and qualify social sustainability, namely, eq-

uity; democracy, participation and civic society; social inclusion and mix; social networking and interaction; livelihood and sense of place; safety and security; and human well-being and quality of life [18].

While social sustainability, as a theoretical concept, includes a multiplicity of values, principles and indicators, these do not provide a rigid framework for applying it to practice. According to Shirazi and Keivani [18], the relevance of social sustainability does not consist of a solid definition that is generally applicable, but of key themes and basic characteristics that should be specified in particular contexts. As Manzi et al. [9] (p. 21), explain: "social sustainability is often more useful as an ambiguous and poorly defined phrase that users can shape to their own circumstances." In short, no universal operational definition to apply social sustainability in cities and neighbourhoods exists. Instead of seeing social sustainability's abundance of aspects that are described in literature as a definition gap that should be filled, we consider the observed ambiguity inherent to what social sustainability conceptually is. This is not problematic in essence—rather, this is a characteristic to be considered when referring to operationalising it in practice.

In this research, we focus on social sustainability as (1) a policy goal in urban planning practices, and (2) its operational form in urban areas. Although research has addressed social sustainability in both functions, little research has focused on how these relate to each other—on how policy goals generate operationalisations and on how operationalisations conform to articulated policy goals. If policy-makers do not succeed in such alignment, they risk outcomes that do not correspond with intended policy goals, or might even oppose to them [24,25].

In order to understand better the relation between goals and operationalisations, we here draw on Moroni's [13] distinction between concepts (i.e., general ideas including some principles that are generally acknowledged) and conceptions, i.e., the diverse, specific forms that the general concepts can adopt. In his perspective, "a *concept* constitutes an abstract ideal on which all participants in a discourse may agree and which can be developed argumentatively in different ways; the realisation and operationalisation of a concept in this sense is achieved by means of a particular *conception*" [13] (p. 9).

More than in concepts, that are relatively little value-laden, the value-laden part mainly appears in conceptions, i.e., during the operationalisation of a concept. Moroni refers to Davy [26] in saying that policy makers either implicitly or explicitly decide on different conceptions when they realize a concept: "This is inevitable. The question is therefore not whether [ . . . ] [the concept] is important for urban policy and planning, but which conception [ . . . ] is chosen for their design" [13] (pp. 9–10). So, if a value-laden concept allows different normative perceptions, we should not pose the question which of them is most true in general, but which of them is most useful for policy.

From this perspective, we can understand social sustainability as a concept that includes a plurality of conceptions and indicators about people's quality of life. These altogether form the criteria of social sustainability as a theoretical concept (see Figure 1). When we shift our focus to its operationalisation, this provides the opportunity to specify its meaning to a specific context of application. In other words, the opportunity to develop a particular conception that is relevant for the issues, problems or questions that policy aims to address. Such conception frames the policy goal. The key point that we aim to stress here is that the conception logically interrelates with the operational form. We see operationalisation as a process of defining operational approaches that support a specific normative conception of a concept. Operational forms thus do not serve as generally valid indicators, but as evaluative tools to the corresponding policy goals. Therefore, if social sustainability is to be applied to a specific policy context, we should not immediately concentrate on its operational indicators. What should first be addressed is what normative conception is regarded most useful for that policy context, and then, what corresponding operational approaches are.

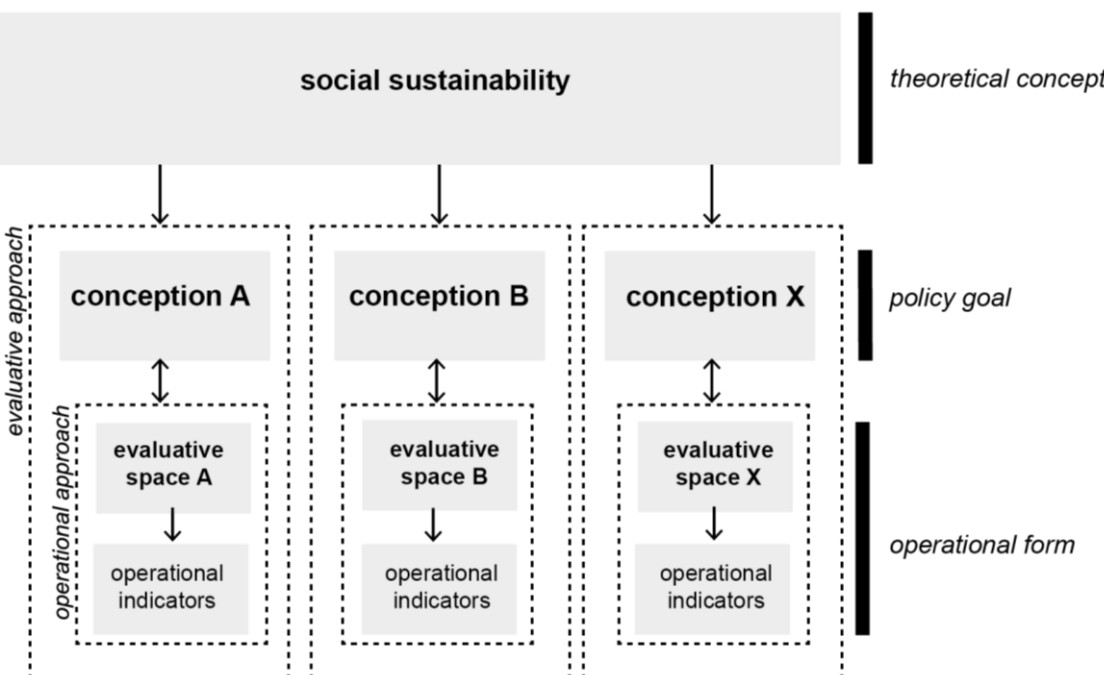

**Figure 1.** Steps of operationalising social sustainability.

Research on social sustainability in the built environment has been concerned with the investigation of operational indicators that relate to social sustainability. Bramley et al. [27] particularly studied the relation between urban form and social sustainability and found among others density, gardens, green space and nearness to bus services to be related to indicators such as safety, friendliness and pride in a neighbourhood. Hamiduddin [28] discusses the relations between demographic compositions, spatial scales and social sustainability aspects. Dempsey et al. [15] listed physical factors of social sustainability, among which are urbanity, decent housing, accessibility and pedestrian friendliness. Similarly, Shirazi and Keivani [29] define density, mixed land use, urban pattern and connectivity, building typology, quality of centre and access to facilities as the "hard infrastructure" of social sustainability in neighbourhoods.

Eizenberg and Jabareen [30] warn that physical indicators presented in studies are akin to general indicators of "good" planning, and that they can have contested effects in cities. For example, interventions to improve walkability in urban areas could lead to more gentrification. They argue that additional features must be added—social sustainability in the built environment is also about processes and social structures in communities "that will emerge within a community and ensure the satisfaction of its needs, which are ever-changing" [30] (p. 3). Accordingly, most studies also include softer, intangible indicators, such as sense of attachment, social networking and interaction [29], social capital and sense of community [15] and local governance structures and inhabitants' perceptions of their influence over their living environment [31]. An overview of the various operational indicators of social sustainability in the built environment has been listed in Table 1.

**Table 1.** Operational indicators of social sustainability in the built environment, based on Dixon and Woodcraft [31], Dempsey et al. [15] and Shirazi and Keivani [29]. See full overview in Table A1 in the Appendix A.

| Tangible | Intangible |
|---|---|
| decent housing | social interaction |
| transport | social networks |
| daily facilities | cultural expression |
| recreation | feeling of belonging |
| jobs | feeling of community |
| schools | safety |
| public spaces | well-being |
| healthcare | existence of informal groups and associations |
| urban design | representation by local governments |
| | levels of participation |
| | levels of influence |

Urban studies have increasingly been including intangible indicators in the social sustainability debate. As Shirazi and Keivani [18] observe, the focus of research has shifted from physical, quantifiable aspects to more qualitative ones, such as sense of place or well-being. This shifting discourse of social sustainability from "hard, traditional themes" to "softer concepts" had already been mentioned by Colantonio and Dixon [10], who pointed out that the shift towards qualitative indicators triggered the debate on what role policy-makers should play in delivering "softer" objectives.

However, they also warned that social sustainability had, until then, not been a serious approach to urban regeneration—opposed to for example cultural industries approaches, health and liveability perspectives and social economy approaches. Whereas such approaches certainly include aspects relating to social sustainability, Colantonio and Dixon argue they do not offer an approach in which social sustainability is a fully integrated dimension of sustainable urban development.

We add to this discussion by pointing at the conceptualization steps that are between understanding social sustainability as a theoretical concept on the one hand, and as an operational indicator on the other hand (see Figure 1). The remaining cloudiness about how to integrate both tangible and intangible aspects of social sustainability in urban planning practices might, in fact, be due to a misfit to an, either explicitly or implicitly applied, conception in policy-making. How much do we know about distinct normative conceptions of social sustainability in planning practices, and what are the options? Is the current search for operational indicators sufficient, or do we need to reinvestigate how distinct conceptions to social sustainability correspond to the various aspects, both tangible and intangible, of social sustainability? In the next section, we take one step back and concentrate on one particular evaluative approach that is promising, yet underexplored, for the understanding social sustainability in urban planning practices, namely, Amartya Sen's [14] capability approach.

## 3. A Capability Conception to Social Sustainability

The core idea of the capability approach (CA) is that in questions of justice or human well-being, one should strive for equality of people's effective opportunities that they have to live a worthy human life [32]. Economist Amartya Sen pioneered the idea on capabilities as a critical, alternative perspective to what equality should be about [33] and developed the CA as an evaluative framework for comparative research on human well-being and quality of life [14]. Philosopher Martha Nussbaum [34] further developed it as a theory of social justice and listed ten non-negotiable capabilities that are central for each person to live a life of dignity. Thereafter, the CA has become an interdisciplinary approach that has been applied in a broad variety of fields, ranging from poverty development [35] to sustainable development [36,37] and education [38]. Moreover, the CA is increasingly

being explored in the urban field. During the recent decade, a quickly growing number of studies has investigated the CA in relation to the built environment [39–41].

The CA focuses on capabilities and functionings as the evaluative space of people's advantage on well-being (Figure 2). Functionings are the doings and beings of a human being, for example traveling, sleeping, being educated and being nourished. Capabilities, then, are the substantive freedoms (i.e., real opportunities) that an individual person has to operationalise these functionings; is a person really able to achieve the functioning that he/she has reason to value [14]? If, for example, traveling is considered as a valuable functioning, the CA poses two evaluative questions: (1) does a person travel? (i.e., evaluating achieved functionings), and (2) if a person does not, could he/she travel if he/she wanted to? (i.e., evaluating capabilities). The CA positions resources as an important factor to achieve capability and functionings, however, it only sees them as means for people to enlarge their capabilities (i.e., ends).

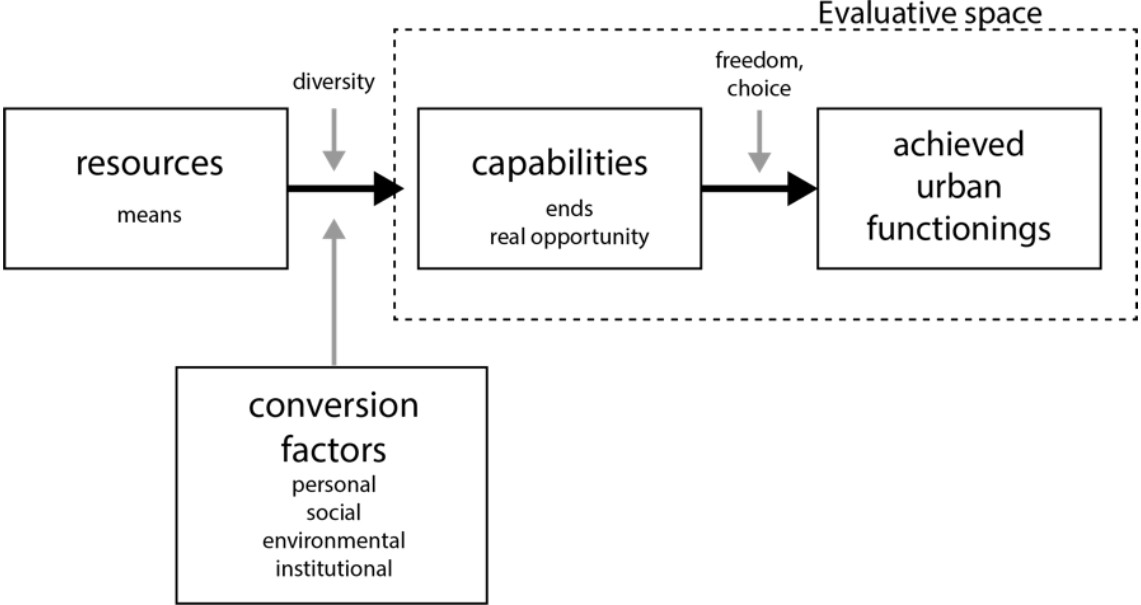

**Figure 2.** A simplified scheme of the capability approach' evaluative framework, based on Sen [14] and Robeyns [42].

According to Sen [14], it is important to emphasize capability as the evaluative space of well-being because human beings are inherently diverse. Focusing on resources as evaluative space would not be fair because equality of resources does not automatically lead to equality of capability. For instance, a disabled person might not have the same access to public transport as an abled person, or two children in the same neighbourhood might not have the same career opportunities because they grow up in different families. An outcome of unequal opportunities despite equal resources is in the CA explained by the context. Sen [14] argues that each person has a unique set of conversion factors that influence how means lead to ends. Conversion factors are personal heterogeneities, environmental diversities, variations in social climate, differences in relational perspectives and distribution within the family. In addition to this, Robeyns [32] distinguishes between personal, social, environmental and institutional conversion factors.

The main contribution of a capability-based conception to social sustainability is that it distinguishes between resources as means and people's real opportunities as ends [14]. Applying a capability-based conception, social sustainability should not be narrowed down to lists of tangible resources because this way would step over the diverse, contextual factors that influence people's eventual capabilities, i.e., people's eventual well-being. The evaluative space of social sustainability should, in the spirit of the CA, focus on actions and possibilities of human beings. Basta [43] suggests to start defining "urban functionings", and opportunities to actually accomplish them in real life. For instance, the social

sustainability indicator "public space" could relate to various functionings and capabilities. From a capability-based operational approach, this indicator could be concerned with the urban functioning "recreating in public space", "making use of public space", or "creating public space" or with the real opportunity to accomplish these functionings. In that way, a capability-based approach broadens the operational definition of social sustainability and considers contextual factors that relate to human well-being before evaluation. As Robeyns [42] (p. 47) puts it, "in order to know what people are able to do and be, we need to analyse the full picture of their resources, and the various conversion factors, or else analyse the functionings and capabilities directly".

The analytical distinction of the CA, between resources and capabilities, allows us to disentangle two evaluative approaches to operationalise social sustainability: a resource-based approach and a capability-based approach. As mentioned earlier, current research on social sustainability stems from an initial focus on tangible, "harder", aspects such as housing, jobs and public space. These aspects refer to indicators as if they are resources in the built environment—the availability of something that a person can make use of. Therefore, an operational approach that focuses on resources as outputs would be valid if the corresponding policy goals are eventually concerned with improvements that concern *the built environment*, such as urban liveability. If policies, however, eventually aim to achieve improvements around *people's actions and opportunities* via the built environment, policy outcomes should rather be assessed in an evaluative space that concentrates on the question whether people *are actually able to make use* of urban resources. A capability-based operational approach would then be a more valid approach.

## 4. Operationalising Social Sustainability in Dutch Urban Planning Practice

So far, we have argued that a capability-based conception has promising advantages to the operational understanding of social sustainability in the built environment. We have also articulated that this conception has been less explored in urban research and practice than a resource-based approach. Next, we will empirically explore our conceptual arguments in the context of urban policy operationalisation in The Netherlands, and question whether a capability-based approach would, in this context, indeed be a better operational approach than a resource-based one.

### 4.1. Emprical Exploration

The exploration is based on desk research of empirical literature on the Dutch planning practice, and conducted as part of an explorative phase in an ongoing study on social sustainability in contemporary urban development projects. It forms the basis for a multiple case-study research on capability-based operationalisations of social sustainability that will be conducted in later phases of the research.

The Netherlands has traditionally often been referred to as a socio-democratic welfare state with a comprehensive integrated planning system, and has moved towards a more liberal approach in the last two decades [44]. Although many years of welfare policies have left their mark in current planning practices, concerning developments about social stability are currently observable in Dutch society [7,8]. We therefore aim to investigate how previous planning practices have operationally been approached, and to what extent these practices have addressed "social sustainability" in its full width. We refer to three dimensions of policy operationalisation in Dutch planning practice that concern social policy goals: (1) national policy programmes for urban renewal, (2) national regulations on the country's social housing system and (3) a national measurement tool on urban liveability.

### 4.2. Three Dimensions of Policy Operationalisation in The Netherlands

The first dimension concerns national policy programmes for urban renewal. The Netherlands has a long tradition in investing in social goals through such programmes. The first large-scale program was operated in the 1970s and mainly consisted out of physical interventions—demolishing neighbourhoods with "slum" dwellings and rebuilding them

with modern housing and public buildings [45]. Later programmes aimed to integrate social goals in urban developments. Two main programmes were Grootstedenbeleid (Big Cities Policies), which aimed at long-term physical, social and economic development in large cities between 1995–2009, and the Krachtwijkenbeleid (40 Neighbourhoods Policy), which aimed to improve forty specific "problematic" neighbourhoods in the Netherlands between 2007–2012 [46,47]. Krachtwijkenbeleid aimed at reducing the number of social housing dwellings in neighbourhoods, replacing rental dwellings to home-owner ones, improving liveability, developing neighbourhood centres, citizen participation and care for citizens with socioeconomic problems [48].

Despite billions invested in these programmes, their impact in urban areas has been contentious. While reports conclude that Grootstedenbeleid has led to visible improvement of neighbourhoods [49], it can be criticized that these improvements have not been substantial enough. Economies had improved, criminality rates had decreased and housing stocks had diversified, but cities still coped with increasing inequality between the "better"- and the "worse"-off citizens and severe social problems among marginalized groups [50]. In addition, Krachtwijkenbeleid has, according to the Social and Cultural Planning Agency (SCP), not led to measurable effects for people's income or for the area's liveability, and even led to a negative effect on the participation of citizens in neighbourhoods [48].

The main point of critique on national policy programmes is that these have focused on the quality of areas, which does not necessarily improve the quality of life of human beings. As Musterd and Ostendof [51] (p. 88) state, "The history of urban policies in The Netherlands can be summarised as follows: a strong focus on area-based approaches in disadvantaged neighbourhoods, aiming to change the housing stock in order to create a social mix". According to the authors, the area-based focus in policies was funded in beliefs that had drifted away from the real situation in practice. In contrast to the policy's aims on social mixing, statistical levels on ethnic- and socioeconomic-segregation levels were in fact not alarming in the Netherlands at that time. Tackling broader structural problems, such as unemployment, would therefore not be effective through area-based initiatives aimed at diversifying housing stocks [51]. In short, the physical rearrangements in urban areas, due to the urban programmes, had little to do with the lives of inhabitants, in contrast to what the programmes aimed to achieve. It is therefore implausible that the urban policy programmes succeeded in comprehensively addressing both tangible and intangible aspects of social sustainability.

The second dimension concerns the use of the country's social housing sector as a tool of social policy operationalisation. The first Dutch housing associations stem from the 1850s, when employers arranged housing for employees and when the "better-off" workers united themselves in housing cooperatives. Hoekstra [52] describes how the social housing sector developed, from these initial forms, into housing associations, led by catholic or protestant initiatives in the beginning of the 20th century, and to an extensive social housing sector, subsidized by the government, between the 1950s and 1990s. In that post-war period, the number of social rent dwellings increased from 10% to 40% of the total Dutch housing stock. The government had strong control and influence on the sector, in order to cope with a large housing shortage. From the 1980s on, however, governmental subsidies disappeared and housing associations became more independent. Nieboer and Gruis mention how the role of housing associations became larger as they became more privatized in the 1990s, and how "the sale of both new and existing homes become more important as a means of financing housing development and as a vehicle for cross-subsidising social activities [i.e., welfare, care, local economy and education]" [53] (p. 278). In this period, housing associations served several societal purposes in neighbourhoods that went far beyond housing provision only.

This has changed, by several reforms, in the last decade. A new housing law, in 2015, prescribed that housing associations should focus on its primary task of housing, called Services of General Economic Interests (SGEI), and that they should transfer all other activities to commercial organisations [52]. This national regulation diminished the capacities of housing associations to engage in broader social activities than housing. So, while housing associations previously fulfilled a role in advancing multiple social sustainability indicators, such as education and well-being, the national policy regulations restrained the social housing sector as a tool to operationalise merely one tangible aspect of social sustainability, namely affordable housing.

The third dimension concerns the Leefbaarometer (Liveability Meter) [54], a state-developed measure instrument that is often referred to in Dutch policy-making discussions about social value in cities. Whereas this instrument has been criticized because it includes some elements that could be perceived as discriminatory [55], it is often applied in neighbourhood studies [49,56] and has become a common tool for urban planning practitioners in order to monitor nonfinancial values in projects [57].

When applying it however, one should not forget that the instrument's aim is to assess people's living environment, and that it does not evaluate people's quality of life [54]. Measuring liveability is not as far-reaching as evaluating social sustainability, as we can observe if we compare indicators of the Leefbaarometer with social sustainability indicators (Table 2). This table shows that the instrument mainly measures tangible indicators of social sustainability, such as the housing stock (e.g., housing quality, typology and tenure), amenities (e.g., proximity to healthcare and schools) and additional indicators about demographics and mutation rates. It does not include indicators that are concerned with social interaction, social networks, feelings of belonging or feelings of community.

**Table 2.** Indicators of social sustainability compared to indicators of the Leefbaarometer. The indicators of the instrument listed here are a summary of the 100 indicators that the Leefbaarometer consists of. See full overview in Table A2 in the Appendix A.

| | Social Sustainability | Leefbaarometer |
|---|---|---|
| **Tangible** | decent housing | housing quality; housing typology; housing tenure |
| | transport | distance to train station and to highway; |
| | daily facilities | number of shops; distance to ATM |
| | recreation | day recreation facilities; number of cafes, restaurants and shops; distance to library; number of stages; distance to swimming pool; proximity to parks and natural areas |
| | jobs | - |
| | schools | number of primary schools |
| | public spaces | - |
| | healthcare | number of general practitioners; distance to hospital |
| | urban design | - |
| | - | demographics |
| | - | mutation rate |
| **Intangible** | social interaction | - |
| | social networks | - |
| | cultural expression | socio-cultural facilities |
| | feeling of belonging | - |
| | feeling of community | - |
| | safety | nuisance; order disturbance; abolishment; violent crimes; robberies; burglaries |
| | well-being | - |
| | existence of informal groups and associations | - |
| | representation by local governments | - |
| | levels of participation | - |
| | levels of influence | - |

## 5. Discussion: Complementarity between Resources and Capabilities

The three dimensions of Dutch policy operationalisation have in common that they mainly address the tangible aspects of social sustainability and scarcely tackle the intangible ones. To wit, the policy programmes for urban renewal were centred around physical, area-based interventions, a new housing law forced the social housing sector to focus merely on affordable housing provision, and the Leefbaarometer mainly focuses on housing and amenity indicators. Although these three examples do not represent The Netherlands' entire urban planning system, they are substantial operational elements of the Dutch urban practice that is concerned with social goals. The examples support the notion in research that intangible aspects of social sustainability have not become as much an integrated approach in urban practices as the tangible ones [10,18].

Our purpose here is not to label operationalisations that focus on area-based, physical interventions as generally ineffective for social policy goals in urban planning. We want to emphasize that physical interventions may contribute to some aspects of social sustainability, such as affordable housing or improved public space, but might by itself not be enough to advance social sustainability in the affected urban areas. As we learn from a vast body of research on neighbourhood effects [58], relations between area-based interventions and human-based improvements are delicate to prove. For instance, Cheshire [59] concludes that studies have not led to ample evidence that living in a poor neighbourhood causes poverty, and that socioeconomically segregated neighbourhoods rather reflect economic inequality than cause it. So, when we evaluate operationalisations of urban policies, these evaluations go hand in hand with the question "what goals do policy interventions pursue?" Do they aim to address tangible goals, such as poverty rates in neighbourhoods, improved urban liveability or changed demographics, or do they aim to achieve more than that?

Currently, a shifting conception of social sustainability goals can be observed in The Netherlands. After a period in which policies have been predominantly operationalised by area-based interventions, more attention is currently called for individual, human-centred perspectives in urban policy-making [60]. Reflecting on the previously applied national policy programmes, Uyterlinde et al. [49] conclude that physical interventions, such as diversifying the housing stock or building new facilities, only add value to neighbourhoods provided that physical conditions are seen as a means to an end. Outcomes of policies should, according to them, eventually be concerned with opportunities of residents, as improving the liveability and safety of neighbourhood should go hand in glove with improving residents' societal opportunities and quality of life [49].

Centralizing area-based goals such as liveability echoes with a resource-based conception to social sustainability, while focusing on human opportunities as outcomes of urban planning interventions complies with a capability-based conception. The two approaches are not dichotomous but rather complementary. A resource-based approach is legitimate because urban planning is a spatial practice that is professionally equipped to create resources in cities relevant to social sustainability, such as housing, schools, libraries, parks, infrastructure or community centres. Resources should not be belittled, also from a capability-conception—how to be educated without a school, or how to enjoy public space without a park? However, the argument of this article is that, by focusing on urban resources as operational indicators, a resource-based approach only addresses social sustainability to a limited extent. It mainly touches upon social sustainability's tangible aspects and therefore steps over many other, potentially unexplored, aspects that are essential for social sustainability.

A capability-based operational approach is complementary to the resource-based approach because it can identify how different groups may have different access to, or make different use of such urban resources. Whereas a resource-based approach seeks for resources as static entities that are generally applicable, a capability-based approach focuses on the relations between human actions and their environment. For instance, the Leefbaarometer's indicator "number of primary schools" informs us about the availability

of this resource in a specific area, but does not tell anything yet about a person's real possibility to send his/her child to a primary school. A new primary school might indeed contribute to increased well-being of local residents; however, it could also be possible that the nearby school has a waiting list for subscription, or that the new school offers a type of education that does not align with the (religious) beliefs of a family. So, although resources can certainly be effective in advancing social sustainability, the question is what other contextual factors affect people's actual opportunity to make use of social sustainability resources. A capability-based approach thus shifts the evaluative space of what should be measured about social sustainability—it is not resources that define levels of social sustainability, but the relations between human beings and these resources.

Although the professional scope of urban planning practitioners is obviously limited and does not allow them to influence all possible contextual aspects that affect a person's capability, the conceptual starting point towards social sustainability makes a difference. Applying a capability-based conception puts the urban planner in a better position to evaluate what role resources and other contextual factors play in achieving social sustainability as perceived by human beings. It makes room for situational flexibility in evaluations and room to specify social sustainability in specific places [18], as it is, according to McClymonth, "a practical approach to judge outcomes and interventions in a range of places and times" [61] (p. 188). Because the capability-based conception centralizes human well-being as the end goal of interferences, this provides the opportunity to go beyond physical, socioeconomic or demographic aspects and to include more aspects that relate to social sustainability. The capability-conception addresses social sustainability from a broader perspective, and therefore, it is more accurate than a resource-based conception.

## 6. Conclusions

The aim of this article was to improve our understanding of the operationalisation of social sustainability in urban planning practices. Our research shows that, between theoretical concept and operational forms, different evaluative approaches towards social sustainability may be taken. The article has argued for one of these—the capability approach—and has shown that, if we want urban areas to become more socially sustainable, it is promising to move from resource-based to capability-based thinking. Our exploration of Dutch policy operationalisations provides some concrete evidence of the gaps that the Capabilities Approach can uncover and fill by focusing on human-centred improvements instead of merely physical, area-centred interventions. Exploring the implications of this approach is promising, because it improves our insight in the factors that influence the way how people use means (i.e., resources like affordable housing, schools and public spaces) for their ends (i.e., capabilities such as the real opportunity to feel part of a community in a neighbourhood). In conclusion, a capability-based conception of social sustainability in cities broadens the operational definition that is currently dominant in urban planning practices, and offers an empirically more accurate definition of what social sustainability is essentially about. This improved understanding can facilitate urban professionals to align operational interventions with their goals around social sustainability, thus, to be more effective in realizing their articulated ambitions.

Complementing resource-based conceptions with capability-based thinking brings social sustainability more in line with the way economic and environmental goals are treated in urban research and practice. In research, it broadens our understanding of social sustainability and explains the diverse ways in which the concept may be applied. For practice, it offers a more comprehensive approach to socially sustainable city planning and acknowledges the context-dependency of its operationalisation in policies and projects.

A risk of taking a capability-based approach to social sustainability operationalisation is that the link between the social dimension and other dimensions of sustainability may be overlooked. The capability approach adopts an anthropocentric world view and identifies human worthiness and dignity as the highest achievable good. Hence, we stress that the objective of making urban areas (more) socially sustainable stems from an overarching

ambition for sustainable development in cities, in which economic, social and environmental dimensions should be equally addressed. This, unavoidably, creates tensions. In practice, urban development projects are vehicles of policy implementation in which various sustainability goals come together and compete, such as decreasing carbon-emissions, generating new jobs or building more affordable housing. Such projects typically span a long period of time. Sustainability goals may fade into the background as the projects are planned, prepared, and executed, either because they are drowned out by other, more dominant policy goals or because they are cancelled due to a lack of funding and/or attention. Next to improving our understanding of how sustainability goals can be comprehensively operationalised, we should thus also create more insight into the ways that goals compete, evolve and are met (or neglected) in real urban projects.

Capability-based conceptions of social sustainability will likely take time and effort to adopt in policy and practice, as it requires in-depth, qualitative inquiry into the differences among the inhabitants of urban areas. However, we hold that a more comprehensive understanding of social sustainability in the built environment will help to identify more, underexplored, factors that affect social sustainability in urban areas. Perhaps the most valuable contribution of applying a capability-based approach is that it opens the floor for discussion on new questions in the social sustainability debate, such as: what do physical interventions in cities aim to achieve, who benefits from them, which inequalities in people's access to resources can we observe, are these inequalities problematic, and what personal, social or institutional factors cause them? Addressing these questions could provide policy-makers with more realistic insights on social sustainability in the built environment and, eventually, with operational tools to pursue more socially stable and vibrant spaces.

**Author Contributions:** Conceptualization, C.J., T.A.D. and C.V.; methodology, C.J. and T.A.D.; formal analysis, C.J.; investigation, C.J.; writing—original draft preparation, C.J.; writing—review and editing, C.J., T.A.D. and C.V.; visualization, C.J.; supervision, T.A.D. and C.V. All authors have read and agreed to the published version of the manuscript.

**Funding:** The research underlying this article is kindly supported by the Stichting Kennis Gebiedsontwikkeling (SKG).

**Data Availability Statement:** Not applicable.

**Conflicts of Interest:** The authors declare no conflict of interest.

## Appendix A

**Table A1.** Operational indicators of social sustainability.

| | Dixon and Woodcraft [31] | Dempsey et al. [15] | Shirazi and Keivani [29] |
|---|---|---|---|
| **Tangible** | | | |
| decent housing | - | decent housing<br>mixed tenure | quality of home<br>building typology<br>social mix |
| transport | transport links | accessibility (e.g., to local services and facilities/employment/green space) | |
| daily facilities | - | - | access to facilities |
| recreation | provision for teenagers and young people<br>shared spaces that enable neighbours to meet<br>space that can be used by local groups | walkable neighbourhood;<br>pedestrian friendly | quality of centre |
| jobs | - | employment | - |

**Table A1.** *Cont*.

| | Dixon and Woodcraft [31] | Dempsey et al. [15] | Shirazi and Keivani [29] |
|---|---|---|---|
| schools | schools | education and training | - |
| public spaces | public space playgrounds | attractive public realm | - |
| healthcare | services for older people healthcare | - | - |
| urban design | - | urbanity local environmental quality and amenity sustainable urban design neighbourhood | quality of neighbourhood density mixed land use urban pattern and connectivity |
| **Intangible** | | | |
| social interaction | how people living in different parts of a neighbourhood relate to each other how well people from different backgrounds co-exist | social interaction social justice social order social cohesion | social networking and interaction |
| social networks | relationships between neighbours and local social networks | social capital social inclusion (and eradication of social exclusion) social networks | social networking and interaction |
| cultural expression | - | cultural traditions | - |
| feeling of belonging | how people feel about their neighbourhood sense of belonging and local identity | sense of community and belonging | sense of attachment |
| feeling of community | - | community cohesion (i.e., cohesion between and among different groups) | - |
| safety | feelings of safety | safety residential stability (vs turnover) | safety and security |
| well-being | quality of life and well-being | health, quality of life and well-being | - |
| existence of informal groups and associations | the existence of informal groups and associations that allow people to make their views known | active community organizations | - |
| representation by local governments | local governance structures responsiveness of local government to local issues | local democracy | - |
| levels of participation | - | participation | participation |
| levels of influence | residents' perceptions of their influence over the wider area and whether they will get involved to tackle wider problems. | - | - |

**Table A2.** Operational indicators of social sustainability compared to indicators of the Leefbaarometer [54].

| Social Sustainability | Leefbaarometer |
|---|---|
| **Tangible** | |
| decent housing | housing part before 1900 |
| | housing part between 1900–1920 |
| | housing part between 1920–1945 |
| | housing part between 1945–1960 |
| | housing part between 1961–1971 |
| | housing part between 1971–1980 |
| | housing part between 1991–2000 |
| | historical housing |
| | dominance of pre-war |
| | dominance of early post-war |
| | dominance of late post-war |
| | dominance of recent buildings |
| | part of single household row-housing |
| | large freestanding and duo-housing |
| | medium-size freestanding and duo-housing |
| | small freestanding and duo-housing |
| | dominance pre-war single household |
| | part of small single household before 1900 |
| | part of small pre-war single household housing |
| | part of small single household housing 1900–1945 |
| | part of small single household housing 1970–1990 |
| | part of small multiple household housing after 1970 |
| | part of single household social rent |
| | part of single household for sale |
| | part of multiple household for sale |
| transport | distance to train station |
| | distance to transfer station |
| | distance to driveway highway |
| daily facilities | number of shops for daily groceries within 1 km |
| | distance to closest atm |
| | day recreation facilities |
| | disappeared supermarket |
| recreation | number of cafes within 1 km |
| | cafes and cafeterias (combined index) |
| | number of restaurants within 1 km |
| | catering industry and shops (combined index) |
| | smaller shops |
| | library within 2 km |
| | number of stages within 10 km |
| | distance to closest swimming pool |
| | proximity to forest |
| | part of green |
| | proximity to parks |
| | proximity to IJsselmeer/Markermeer |
| | proximity to recreative water |
| | proximity to North Sea coast |
| | proximity to North Sea |
| jobs | - |
| schools | number of primary schools within 1 km |
| | education and healthcare (combined index) |
| public spaces | - |
| healthcare | number of general practitioners within 3 km |
| | distance to closest hospital |

**Table A2.** *Cont.*

| Social Sustainability | Leefbaarometer |
| --- | --- |
| urban design | urban facilities<br>part of national monuments<br>part of buildings with industrial function<br>part of buildings with public function<br>density<br>proximity to residential area<br>proximity to 'open, dry, natural area'<br>water in neighbourhood<br>high voltage pylonsnoise pollution<br>distance to main road network<br>distance to high way<br>number of trains<br>proximity to rail track<br>proximity to roads<br>proximity to chloride area<br>industry nearby<br>flood risk<br>earthquake risk |
| - | mutation rate |
| - | part of wester migrants<br>part of 'moe-landers'<br>part of non-western migrants<br>part of Moroccans<br>part of Surinamese<br>part of Turks<br>part of other non-western migrants<br>single parent families<br>families with children<br>families without children<br>part of incapacitated<br>part of welfare recipients<br>elderly<br>development of households<br>development of 15–24 year old's |
| **Intangible** | |
| social interaction | socio-cultural facilities |
| social networks | - |
| cultural expression | - |
| feeling of belonging | - |
| feeling of community | - |
| safety | nuisance (combined index)<br>order disturbance<br>abolishment<br>violent crimes<br>robberies<br>burglaries |
| well-being | - |
| existence of informal groups and associations | - |
| representation by local governments | - |
| levels of participation | - |
| levels of influence | - |

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
