# Peer review of "Planning for Urban Social Sustainability: Towards a Human-Centred Operational Approach"

_sustainability, doi:10.3390/su13169083_

Round 1

Reviewer 1 Report

An interesting base for the further case-study research.

Just minor suggestion that the Table 1 and 2 as well as the extended version of Table 1 in the Appendix, could be set up a bit more aesthetically (maybe smaller font size, a gap added between the text and vertical line-break between columns).

Reviewer 2 Report

The conceptual bases of the article and its methodology are really very well raised. The introduction is pleasantly explanatory in relation to the problems of conceptualization of the so-called social sustainability. The authors’ discussion of the different conceptualizations of this often ambiguous concept is also very valuable. All the authors referenced in the article are relevant, including the philosopher Martha Nussbaum, who lists the core capabilities that should be supported by democratic countries.

One of the problems with intangible indicators is their objectification or, in other words, their objective measurement. The authors of the article acknowledge that capability-based conceptions of social sustainability will take time and effort to adopt in policy and practice, as it requires in-depth, qualitative inquiry into the differences among the inhabitants of urban areas.

One might wonder if the operationalization of social sustainability goals are universally applicable, or are only valid for western cities such as the Netherlands.

It is not necessary that aspects such as those mentioned be considered or added to the article.  We hope this article serves to emphasize the need to include a capability-based operational approaches in the urban planning policies of our cities.

Reviewer 3 Report

Well written and very interesting paper.

This research is both relevant to today's social climate and addresses Social Sustainability which is often overlooked in research on urban sustainability. It is an interesting paper that flows well with clear text that makes it easy to read.  

The topic is quite original and adds the intangible aspects to the more common, tangible urban sustainability arguments by explaining the capabilities approach.

The authors' conclusion of the necessity to move from a resource-based approach to a capability-based approach in order to make urban areas more socially sustainable is supported by the evidence provided. 
